# Biogenic Silver Nanoparticles from *Iris tuberosa* as Potential Preservative in Cosmetic Products

**DOI:** 10.3390/molecules26154696

**Published:** 2021-08-03

**Authors:** Maria Mondéjar-López, Alberto José López-Jiménez, Minerva Abad-Jordá, Angela Rubio-Moraga, Oussama Ahrazem, Loudes Gómez-Gómez, Enrique Niza

**Affiliations:** 1Instituto Botánico, Universidad de Castilla-La Mancha, Campus Universitario s/n, 02071 Albacete, Spain; maria.mondejar3@alu.uclm.es (M.M.-L.); albertojose.lopez@uclm.es (A.J.L.-J.); angela.rubio@uclm.es (A.R.-M.); Oussama.ahrazem@uclm.es (O.A.); MariaLourdes.gomez@uclm.es (L.G.-G.); 2Nirvel Cosmetics, S.L, Polígono Industrial Costes Baixes, Carrer C, 9, 03804 Alcoi, Spain; mabadjorda@gmail.com

**Keywords:** nanotechnology, silver nanoparticles, antimicrobial, biotechnology, phytochemistry, plant extract, green chemistry, cosmetics, preservatives

## Abstract

Biogenic-silver nanoparticles emerge as new nanosilver platforms that allow us to obtain silver nanoparticles via “green chemistry”. In our study, biogenic-silver nanoparticles were obtained from *Iris tuberosa* leaf extract. Nanoparticles were characterized by a UV-vis spectroscopy, dynamical light scattering technique. The transmission electron microscope revealed spheric and irregular nanoparticles with 5 to 50 nm in diameter. Antimicrobial properties were evaluated against typical microbial contaminants found in cosmetic products, showing high antimicrobial properties. Furthermore, natural moisturizing cream was formulated with biogenic-silver nanoparticles to evaluate the preservative efficiency through a challenge test, indicating its promising use as preservative in cosmetics.

## 1. Introduction

The global cosmetic market is projected to register a CAGR (compound annual growth rate) of 4.3% during the forecast period (2016–2022) and is anticipated to reach $429.8 billion by 2022 [1]. This rising market needs to place special interest in multidimensional control to oversee toxic ingredients and microbial contamination [2]. The contamination of cosmetic products (CPs) is one of the most problematic risks for consumer health. During the period between 2008 and 2014, the Rapid Alert System (RAPEX) of the European Commission (EC) notified 62 cases of CPs contaminated by microorganisms that were recalled due to these incidences [2].

In general terms, the most common alterations in CPs are through microorganism contamination or the result of over exposition to atmospheric oxygen [3]. Every time a new CP is opened, the microorganisms that are present in the atmosphere come in contact with the product. However, those products that have the appropriate preservatives determine the development of these microorganisms, allowing the product to remain safe. Usually, the origin of contamination in new CPs comes from one or more sources, such as raw material, environment, equipment used during manufacture, primary packaging material, or handling by personnel. When microorganisms come into contact with an inadequately preserved cosmetic product, this can be affected in different ways that include the appearance of mold on the product, separation of phases of the emulsions, loss of viscosity, change in the aroma, or rancidity of fats [2]. Although the risk of developing disease from CPs is very low, the bacteria present in these products can cause irritations or infections, especially if the product comes into contact with broken skin. About 2 to 4% of dermatological consultations are due to dermatitis caused by cosmetics. In addition, adverse reactions affect not only the skin in the form of irritation or peeling but also as conjunctivitis, asthma, urticaria, angioedema, or pneumonia [4]. To avoid these problems, substances or additives providing stability are included in the formulation of cosmetic products. There are different forms of conservation that can range from physical, radiation and chemical conservation [5].

The preservative systems most used in cosmetics are natural antioxidants (Vitamin E) and synthetic ones (sodium bisulfite) as well as antimicrobials and antifungals such as parahydroxybenzoates (Parabens) or formaldehyde-releasing agents. An effective preservative should have a broad activity spectrum and a longer duration than the cosmetic product itself, being equivalent to the expected shelf-life plus the usage time [6].

Some cosmetic preservatives such as Parabens, Triclosan, Benzalkonium chloride, Imidazolidinyl urea, and Diazolidinyl urea have shown different adverse effects in humans such as DNA damage [7], antiandrogenic activity [8], estrogenicity [9], endocrine disruptors [10,11], cytotoxic and genotoxic effects on human lymphocytes, risk of cancer in humans, allergic reactions, reproductive disorders, and environmental and animal toxicity [1,11,12,13]. For these reasons, some of the traditional preservative systems are currently being questioned, so new molecules with biocidal activity or molecules that create an unfavorable environment for microorganisms are being sought.

Nanotechnology and its “nanomaterials” (NMs) arise as new and promising alternatives to traditional and hazardous ingredients in CPs owing to its physicochemical properties [14]. In the EU, the official definition of NMs in cosmetics is given as: “an insoluble or bio-persistent and intentionally manufactured material with one or more external dimensions, or an internal structure, on the scale from 1 to 100 nm”. There are a lot of types of NMs that are currently used in CPs, and most of them are registered in different regulatory organizations. In the EU, the main regulatory framework for cosmetic products is the EC Regulation 1223/2009, which includes a list of all NMs used in CPs [15]. The last updated catalogue consisted of 29 NMs.

Silver nanoparticles (AgNPs) are promising NMs used for medical, food, health care, and consumer and industrial purposes, due to their unique physical and chemical properties, and especially for their behavior as a biocide [16]. Within the NMs most used in cosmetics, colloidal silver (nano) or (AgNPs) are widely used as a chemopreventive agent in sunscreens and as a preservative agent in CPs. It is considered as a safe preservative by the Scientific Committee on Consumer Safety (SCCS) [17].

There are three main approaches for obtaining AgNPs: chemical, physical, and biological [18]. Biological methods are a new ecological alternative that does not employ toxic or non-environmental friendly reagents in their synthesis [19]. This approach utilizes natural resources such as plant extracts, bacteria, fungi, and other templates (DNA, viruses) as a source of reducing agents for the generation of AgNPs when they react with silver molecules under different conditions [20,21].

There are different biogenic silver nanoparticles available, which are made from several plant extracts such as green tea, *Salvia lerifolia, Psidium guayaba, Nebumbo nucifera,* and *Emblica officinalis,* among many others. These nanoparticles have been tested and used successfully as biocide agents [22].

*Iris tuberosa,* mainly known as snake’s head, widow iris, black iris, or velvet flower-de-luce, is a species of non-rhizomatous plants belonging to the Iridaceae family. *Iris tuberosa* has been used in traditional medicine for acne treatment, for skin care, and for dandruff treatment.

The aim of this work is to formulate a new biogenic-AgNP to be used as a preservative in moisturizing cream. The water leaf extract of *Iris tuberosa* was used as a precursor of reducing agents to obtain bio-AgNPs. The AgNPs were characterized and evaluated using UV-Vis spectrum, dynamic light scattering (DLS) technique and transmission electron microscope (TEM). Furthermore, the antimicrobial and antioxidant properties were determined, and finally a moisturizing cream incorporating our biogeninc-AgNps was formulated, and its efficacy and efficiency were evaluated as preservative agents using a challenge test.

## 2. Results and Discussion

### 2.1. Polyphenol and Flavonoid Contents in Iris tuberosa Extract (ITE)

Polyphenols and flavonoids are among the most commonly used agents to produce AgNPs by green chemistry through plant extracts [22,23]. These compounds are used as reducing agents to produce silver nanoparticles reacting with a silver precursor. Many silver nanoparticles have been successfully obtained using plant extracts, especially those with a high content of polyphenols and flavonoids. In this study, an aqueous extract from *Iris tuberosa* (ITE) was obtained and the total content of polyphenols and flavonoids were quantified using the Folin-Ciocalteau method and colorimetric method with AlCl_3_*6H_2_O, respectively. The results showed that the ITE extract contained an amount of total polyphenols of 1305 mg of GAE/100 gr of dried weight and a total flavonoid content of 1966 mg of QE/100 gr of dried weight. Other extracts used as precursors of silver nanoparticles contained these compounds, e.g., an aqueous extract of pomegranate leaves, where AgNPs were obtained, had a polyphenol total of 397 mg GAE/gr and a flavonoid total of 126 RE/gr [23].

### 2.2. Synthesis and Characterization of Biogenic-AgNPs

#### 2.2.1. Synthesis and UV-vis Characterization of Biogenic-AgNP

Silver nanoparticles were biosynthesized using ITE aqueous extract. Average total polyphenol and flavonoids content was 13.05 mg of GAE/g and 19.66 mg of QE/g of dried weight, respectively. As soon as the AgNO_3_ solution was added to the aqueous leaf extract from ITE under vigorous stirring, the silver nitrate solution color immediately turned from light yellow to brownish yellow, indicating the formation of silver nanoparticles as depicted in Figure 1. The brownish yellow color was due to the excitation of free electrons in the nanoparticles and occurred in 60 min at room temperature and under white light. After 24 h, no further color change was detected due to stabilization of the synthesized nanoparticles.

To evaluate the biosynthesized nanoparticles of AgNPs through green synthesis using the aqueous ITE extract, a UV-visible scan was done from 380 to 500 nm, in order to confirm the presence of nanoparticles in the sample based on their optical absorbance peaks. The absorption spectra of the synthesized silver nanoparticles was recorded against water. The reduction of silver ions was confirmed by UV-vis spectra, where the maximum absorbance was seen at 454 nm (Figure 1). The characteristic peak of silver nanoparticles known as the surface plasmon resonance increased in a timely manner and remained in the range of 420–480 nm, as shown in silver nanoparticles obtained from *Lysiloma acapulcensis* [24], and with an *Althernantera sessilis* extract, where similar peaks were obtained [25]. ITE extract acts as a reductant agent mediating the synthesis, as well as stabilization, of the silver nanoparticles with a characteristic plasmon resonance band in their characteristic range.

#### 2.2.2. DLS Measures of Biogenic-AgNPs

The hydrodynamic diameters of AgNPs in aqueous solution were determined using dynamical light scattering. DLS or photonic correlation spectroscopy is a spectroscopy technique that allows for the evaluation of nanoparticle size and surface charge of the stern layer through Brownian movement [26]. The main advantage of DLS is the short time required to perform measurement, and the relatively low cost of the apparatus; therefore, it is becoming the preferred method for nanoparticle sizing. However, the DLS method has several drawbacks with respect to the influence of dust particles or small amounts of large aggregates. The results listed in Table 1 and Figure 2c show that the nanoscale of biogenic-AgNPs was 116 nm with a polydispersity index (PDI) of 0.3, due to a range of nanoparticles between 24 nm to 105 nm. The higher sizes detected are the consequence of aggregation of smaller nanoparticles, which lead to a high value of PDI. Most literature indicates a similar particle size of AgNPs ranging from 10–800 nm as described in neem, onion, and tomato extracts with a PDI between 0.2 and 1.0 [27]. Similar results were obtained using a *Berberis* leaf extract where the silver nanoparticles showed a size between 90 and 100 nm, and a PDI of 0.3. The authors of the study point out that although these nanoparticles had high DL values, the aggregation of small nanoparticles was detected by TEM analysis [28].

Zeta potential determination allows or the estimation of the surface charge, which can be used for checking the physical stability of nanoparticles [29]. A large positive or negative value of zeta potential of nanoparticles point out good physical stability of nanoparticles, due to electrostatic repulsion of individual particles. Biogenic-AgNPs from ITE in Table 1 and Figure 2d showed a Z-value of −27.5 mV, indicating stable and monodisperse suspensions [30]. A zeta potential value other than −30 mV to +30 mV is generally considered to have sufficient repulsive force to attain better physical colloidal stability. A small zeta potential value can result in particle aggregation and flocculation due to the Van der Waals attractive forces acting upon them.

#### 2.2.3. TEM Images of Biogenic-AgNPs

To gain more information on the obtained nanoparticles, their size and morphology were analysed by transmission electron microscope (TEM). TEM micrographs shown in Figure 2a,b revealed spherical and irregular shapes. The micrograph shows nanoparticles with a maximum size of 50 nm and a minimum of approximately 5 nm, which was in concordance with the data obtained in DLS measurement, thus indicating that the high value is due to aggregation of small nanoparticles. Smaller-sized Ag nanoparticles have many positive characteristics, such as chemical stability; good conductivity; cancer treatment; antibacterial activities; and antifungal, antiviral, and biofilm eradication, which would make them attractive for many industrial applications.

### 2.3. Antimicrobial Activity of Biogenic-AgNPs

It is generally assumed that the attachment of AgNPs onto the cell wall and membrane plus the intracellular damages caused by AgNPs and silver ions leads to oxidative stress. Augmentation of the concentration of reactive oxygen species (ROS) can be attributed to a higher rate of formation or to a disruption in the scavenging pathways. These mechanisms either singly or concurrently are involved in the antibacterial actions and lead to cellular inactivation.

In vitro evaluation of antimicrobial properties of biogenic silver nanoparticles against typical pathogen contaminants in cosmetics, i.e., *Escherochia coli*, *Pseudomonas aeruginosa*, *Staphylococcus aureus*, *Candida. Albicans*, and *Aspergillus brasiliensis,* was realized by the broth microdilution method. Data from Table 2 indicates that the AgNP has a high MIC ranging from 14.1 μg/mL against *P. aeruginosa*, *S. aureus*, and *A. brasiliensis* to 66.7 μg/mL in *C. albicans*, while the control MIC was lower ranging between 185 μg/mL against *C. albicans* and *A. brasiliensis* and 3333 μg/mL in *S. aureus*. Silver nanoparticles are considered the most widely explored antibacterial nanoagent due to their broad-spectrum antimicrobial properties and robust antimicrobial effectiveness against bacteria, viruses, and fungi. AgNP acts as antimicrobial agent mainly through three mechanisms: (I) cell wall and membrane damage, inhibiting the respiratory process through interaction between silver and compounds of thiol groups [28]; (II) intracellular penetration and production of DNA and protein damage [31]; and (III) oxidative stress through ROS, producing membrane breakage and increasing membrane permeability, which finally results in disruption of the electron transport chain and leakage of the cellular content [22]. Table 3 shows similar MICs between gram − and gram + species, revealing that the antimicrobial activity is unrelated to the class of bacteria. However, other AgNPs synthesized by *Berberis vulgaris* extract showed high affinity for gram negative species such as *E. coli* than for gram positive species as *S. aureus* [28].

Biogenic-AgNPs shows a lower MICs against *P. aeruginosa* and *S. aureus* than against *E. coli* with 14.1 µg/mL and 44.4 µg/mL, respectively. Other reports using silver nanoparticles from *Alpia katsumadai* showed MICs of 20 µg/mL against *S. aureus* and *E. coli* and 40 µg/mL against *P. aeruginosa* [32]. More effective inhibition was obtained by silver nanoparticles from *Lotus* extract showing an MICs against *P. aeruginosa* and *S. aureus* of 10 µg/mL [33]. Aragão et al. showed an MICs for *E. coli* and *S. aureus* of 34.3 and 81.2 µg/mL, respectively, with silver nanoparticles made from *Gracilaria birdiae*. These results confirm that our biogenic-AgNPs have a good antibacterial activity similar to those reported in other studies. *C. albicans* was less sensitive to Biogenic-AgNPs with an MICs of 66.7, while *A. brasiliensis* had an MIC of 14.1 µg/mL, exhibiting reasonably strong antifungal activity (Table 2). Such results are in a good agreement with the previously published paper considering silver nanoparticles from *Enteromorpha flexuosa* and *A. brasiliensis* showing MICs against *C. albicans* ranging between 25 and 300 µg/mL, repectively [34,35].

### 2.4. Antioxidant Properties of Biogenic-AgNPs

Nowadays, there is a great interest in products with antioxidant activity. These products are frequently used as food additives and cosmetics and are regulated by law;of special interest are food antioxidants in the prevention of inflammatory lesions, nutritional deficiencies, autoimmune diseases, Parkinson’s, heart attacks, myocardium, aging, neoplasms, neurodegeneration, atherosclerosis, and diabetes. The antioxidant activity of a compound is due to the capture of free radicals and ROS, produced by cellular metabolism or in response to external factors that are inactivated, thus avoiding or preventing degenerative diseases in humans caused by oxidation of nucleic acids, proteins, or lipids. In order to check the possible antioxidant activity of these nanoparticles and expand their uses, we evaluated these properties with the DPPH scavenging method. DPPH was chosen for its simplicity, it is one of the few organic radicals with nitrogen atoms in its structure, which confers the stability as a result of the delocalization of an unpaired electron on the molecule. This delocalization also causes an intensification in the purplish color characteristic of the radical, which in a metallonic medium absorbs at 515 nm. This purplish coloration of the solution is attenuated in the presence of an antioxidant that can donate or transfer a hydrogen atom, giving a yellowish color due to the reduced form of DPHP-H [36]. Biogenic-AgNPs exhibited a DPPH radical scavenging reduction (%) from 24–65% at maximum dose of 1.4 mg/mL, as depicted in Figure 3. It can be therefore concluded that the biogenic-AgNPs formed displayed an antioxidant activity similar to silver nanoparticles formulated from *Alternathera sessilis* having a maximum of DPPH scavenging reduction of 62% [25].

### 2.5. Preservative Efficacy of Biogenic-AgNPs into Formulated Cream

The cosmetic industry was one of the first to use and develop patent applications using nanotechnology and nanomaterials. Applications cover product formulation, packaging, as well as cosmetic manufacturing equipment. In cosmetic products, nanomaterials are used as active carrier substances and/or as formulation support in order to improve the effectiveness of the product. In order to evaluate the use of biogenic-AgNPs as a preservative agent, a cream was formulated as displayed in Table 3.

Following the UNE-EN ISO 11930, control and AgNPs-based creams were inoculated separately with *E. coli*, *P. aeruginosa*, *S. aureus*, *C. Albicans,* and *A. brasiliensis* to evaluate the preservative efficacy through a challenge test. After 2, 7, 14, and 28 days, one gram of both creams was spread with a drigalski spatula in petri dishes containing a PDA for fungi and LB for bacteria to count CFUs. The results of the challenge test are shown in Table 4. The UNE-EN ISO 11930 established two types of acceptance criteria: (A) for bacteria, the preservative should reduce the number of bacteria to 2 logs on day 2, 3 logs on day 7, and no CFU on day 28; for fungi, the policy dictates a reduction of 2 logs on day 2 and no CFU on day 28; (B) for bacteria and fungi, the preservative should inhibit the growth to 3 logs and 1 log on day 14, respectively, and no CFU on day 28 in both microorganisms. Based on these criteria and the results shown in Table 4, our biogenic-AgNPs are an effective preservative system satisfying the A and B criteria of acceptance of the mentioned normative. The control cream was formulated using benzyl alcohol as a mild preservative agent, which is obtained by sugar fermentation and is commonly used in natural cosmetics due to its safety profile [37], while an ethylhexylglycerin is a multifunctional agent commonly used as a natural preservative and skin conditioning agent [38]. The cream control had less of a preservative effect than Biogenic AgNP cream. Thus, our proposed formulation confirmed that biogenic-AgNPs are more effective preservatives for cosmetics than common “eco-friendly” preservatives.

## 3. Materials and Methods

All the reagents were purchased from Sigma Aldrich (Madrid, Spain). Plants of *Irirs tuberosa* were collected in the Botanical Garden of Castilla-La Mancha (Albacete, Spain). Microoganisms were purchased from the American Type Culture Collection (Manassas, VA, USA) *E. coli* (ATCC25922), *P. aureginosa* (ATCC27853), *S. aureus* (ATCC 6538), *C. albicans* (ATCC 10231), and *A. brasiliensis* (ATCC16404).

### 3.1. Plant Extract Preparation

*Iris tuberosa* leaves were pulverized and lyophilized. Then, one gram of these lyophilized leaves was added to 100 mL of milliQ water and kept on a heated plate for 15 min at 80 °C. The aqueous extract was filtered through 0.45 µM millipore filter and stored at −20 °C.

### 3.2. Content of Flavonoids and Polyphenols in Iris tuberosa Extract

**Determination of total polyphenols.** The total amount of polyphenols in the aqueous extract was determined according to the Folin-Ciocalteau method [39]. Briefly, 0.1 mL of aqueous extract was mixed with 2 mL of 2% Na_2_CO_3_, 2.8 mL of H_2_O and 0.1 mL of Folin-Ciocalteau reagent. After mixing, the color change was measured by the absorbance at 750 nm. The calibration curve was done with Gallic acid as standard at different concentrations (10–200 ppm). The experiment was carried out in triplicate.

**Determination of total flavonoid content.** The total amount of Flavonoids in the aqueous extract was determined through a colorimetric method with AlCl_3_·6H_2_O [39]. Briefly, 0.5 mL of aqueous extract was mixed with 1.5 mL of ethanol, 0.1 mL of AlCl_3_·6H_2_O at 10%, 0.1 mL of CH_3_COOK 1 M, and 2.8 mL of H_2_O. After mixing, the color change was measured at 415 nm. The calibration curve was realized with quercetin as standard at different concentrations (8–500 ppm). The experiment was realized in triplicate.

### 3.3. Synthesis of Biogenic-Silver Nanoparticles

**Synthesis of AgNPs.** Following a method previously described [24], with several modifications, 10 mL of AgNO_3_ 25 mM was added dropwise at a caudal of 2 mL/min into 10 mL of *I. tuberosa* aqueous extract under vigorous stirring. The suspension was kept in agitation under white light, monitoring color change of the suspension over time. The AgNPs were collected after centrifugation at 15,000 rpm for 15 min at 4 °C and washed several times with milliQ water and freeze dried at −40 °C.

### 3.4. Nanoparticles Characterization

**UV-vis spectra analysis.** The AgNPs formation was performed by measuring the UV-vis spectrum of the reaction mixture against *I. tuberosa* extract as a blank. The spectral analysis was done using a double beam PerkinElmer spectrophotometer at a resolution of 1 nm from 380 nm to 500 nm.

**Particle size measurements.** The average sizes, polydispersities, and Z-potentials of the AgNPs were measured using a Zetasizer Nano ZS (Malvern Instruments). Data was analyzed using the multimodal number distribution software included in the instrument. All measures were done in triplicate.

**Morphology studies of AgNPs.** High resolution electron microscope images were obtained on a Jeol JEM 210 TEM microscope operating at 200 kV and equipped with an Oxford Link EDS detector. The resulting images were analyzed using Digital Micrograph™ software from Gatan.

### 3.5. Antimicrobial Assay

The antimicrobial activity and minimum inhibitory concentration (MIC) of biogenic-AgNPs were tested against the most common contaminants in cosmetics, and those that UNI EN ISO 11930:2012 recommend for preservative efficacy evaluation. A challenge test using *P. aeruginosa, E. coli, S.s aureus, A. Brasiliensis,* and *C. albicans* was carried out using the broth microdilution method [40]. Stock cultures were prepared from Culti-Loops ™ (Sigma-Aldrich, Madrid, Spain) in Nutrient Broth (NB) and Potato Dextrose Broth (PDB) at 37 °C. Standarized inoculum was then created by dilution in Müller Hinton medium to a final density of 0.5 McFarland units by densitometer McFarland type DEN-1B (Biosan, Riga, Latvia). AgNPs were tested in concentrations of 133 µg/mL to 0.5 µg/mL. Gentamicin (for bacteria) and Tebuconazole (for mold and yeast) were used as standards. After treatment, plates were incubated 24 h at 37 °C for bacteria and 48 h at 30 °C for yeast and fungi.

### 3.6. 2,2-Diphenyl-1-Picrylhydrazyl (DPPH) Radical Scavenging Activity

FRS, free radical scavenging activity, was determined as described previously [41], and 0.5 mL for each concentration (1.5 mg/mL, 750 µg/mL, 375 µg/mL, 187.5 µg/mL, and 93.8 µg/mL) of synthetized AgNPs was mixed with 0.2 mM methanolic DPPH radical solution (0.5 mL). Equal volumes (1 mL) of AgNPs and DPPH solution (0.2 mM in ethanol) were mixed and kept in the dark at room temperature for 30 min. The absorbance of the solution was measured at 517 nm. The FRS was calculated by % = (A0 − A1/A0) × 100.

### 3.7. Moisturizing Cream Formulation

Two types of moisturizing creams were formulated: Control Cream with standard preservatives (Benzyl alcohol + Ethylhexylglycerin) and AgNP-Cream preservative-free with the addition of biogenic-AgNPs. The composition of both creams is shown in Table 4. Oil phase (B) and water phase (A) were preheated at 70 °C to achieve the fusion of oils and waxes present in this phase. Then, B was added slowly under agitation in a homogenizer to form an oil/water emulsion. Upon cooling, the thermolabile compounds (C), preservatives in the case of the control cream and biogenic-AgNPs, at a dose of 70 µg/mL (Maximum of MIC value), in the AgNP-cream, were added to the cooled cream under continuous agitation.

### 3.8. Preservative Activity of AgNPs in Formulated Cream

The preservative efficacy of AgNPs in the moisturizing creams was carried out following the criteria of UNE-EN ISO 11930. Twenty grams of each cream (Control and AgNP) was diluted with a sterile NaCl solution at 0.9%. Then, each cream was inoculated with 10^5^ UFC/mL for bacteria, 10^4^ CFU/mL for yeast, and 10^3^ CFU/mL for mold. Contaminated creams were stored at room temperature for 30 days. After 2, 7, 14, and 28 days, one gram of each contaminated cream was diluted and spread in Petri dishes to count CFUs of bacteria. Yeast and mold contaminations were evaluated at 7, 14, and 28 days using the same method.

## 4. Conclusions

The incorporation of preservatives in cosmetic formulations is necessary because these products are a source of nutrients for bacteria, fungi, and yeasts. However, finding the right type of preservative or preservative system to incorporate into each formula, which satisfies all preservation and toxicological safety criteria, represents a challenge for the cosmetic microbiologist. Our Biogenic AgNPs preservative showed a broad spectrum of antimicrobial activity, has a known chemical structure, is completely soluble in water, is compatible with all the ingredients of the formulation, and is inexpensive to produce. Therefore, it can be easily transferred to the cosmetic industry.

## Figures and Tables

**Figure 1 molecules-26-04696-f001:**
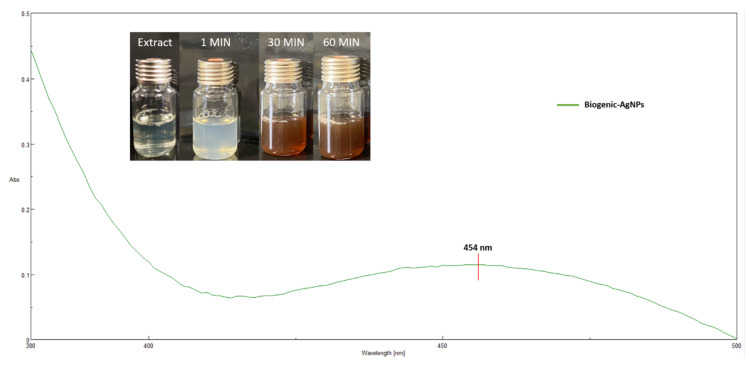
UV-spectrum of *Iris tuberosa* extract and Biogenic-AgNPs. Inset shows the change in color of the silver nanoparticle formation.

**Figure 2 molecules-26-04696-f002:**
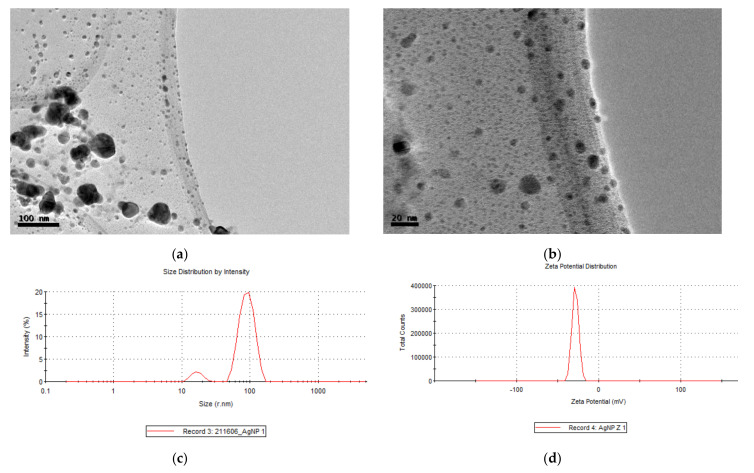
Nanoparticle characterization by microscopy and Z potential. (**a**) and (**b**) TEM micrographs, (**c**) size distribution, and (**d**) Z potential of Biogenic-AgNPs.

**Figure 3 molecules-26-04696-f003:**
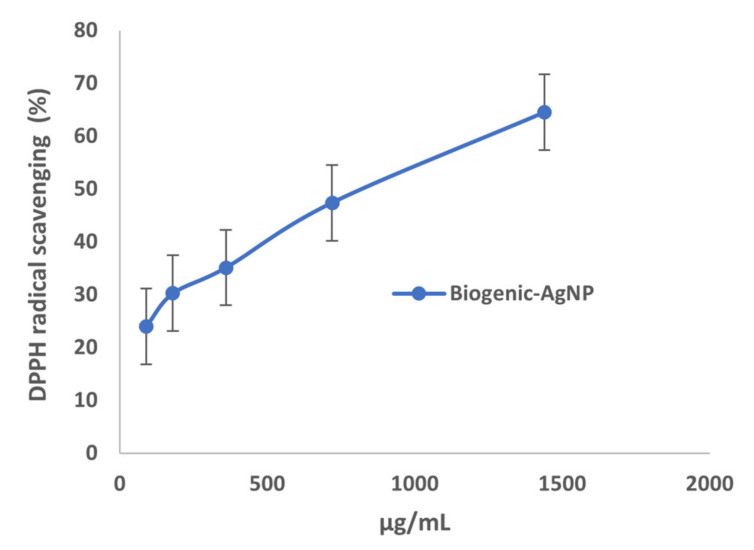
DPPH scavenging reduction of Biogenic-AgNPs.

**Table 1 molecules-26-04696-t001:** DLS measures of Biogenic-AgNPs.

Formulation	Average Size (nm)	PDI	Z-Value (mV)
Biogenic-AgNPs	116.4 ± 4.10	0.3 ± 0.02	−27.5 ± 0.83

**Table 2 molecules-26-04696-t002:** Minimum inhibitory concentrations (µg/mL) of control drugs (Gentamicin (*) and Tebuconazole (**)) and Biogenic-AgNPs.

Microorganism	Control MIC (µg/mL)	Biogenic-AgNP MIC (µg/mL)
*E. coli*	* 1111.0 ± 7.3	44.4 ± 2.5
*P. aeruginosa*	* 1111.0 ± 7.8	14.1 ± 2.6
*S. aureus*	* 3333.0 ± 19.1	14.1 ± 2.7
*C. albicans*	** 185.0 ± 7.6	66.7 ± 0.4
*A. brasiliensis*	** 185.0 ± 9.8	14.1 ± 0.8

**Table 3 molecules-26-04696-t003:** Cream composition. A (Aqueous phase), B (Oil phase), and C (thermolabile compounds).

Ingredient	Control-Cream (%)	AgNP-Cream (%)
Water (A)	56.3	57.3
Vegetable glycerin (A)	4.2	4.2
Macadamia ternifolia seed oil (B)	12.5	12.5
Orbingnya oleifera seed oil (B)	10	10
Helianthus annus seed oil (B)	5	5
Montanov 68 (B)	5	5
Argania spinosa kernel oil (B)	2	2
Biosaccharide gum-1 ©	2	2
Malva silvestris flower extract ©	1.27	1.27
Vitamin E (C)	0.5	0.5
Parfum (C)	0.2	0.2
Benzyl alcohol (C)	0.91	-
Ethylhexylglycerin (C)	0.12	-
AgNPs (C)	-	0.007

**Table 4 molecules-26-04696-t004:** Challenge test in formulated Control-Cream and AgNP-Cream.

Microorganism	Time (Days)	Control-Cream	AgNP-Cream
*P. aeruginosa* Count CFU/gr	0	1.5 × 10^5^ ± 16,773.0	1.5 × 10^5^ ± 7810.2
	2	1.5 × 10^5^ ± 2098.4	0
	7	1.5 × 10^5^ ± 20,207.3	0
	14	1.5 × 10^5^ ± 1000	0
	28	1.2 × 10^4^ ± 321.5	0
*E. coli* Count CFU/gr	0	1.5 × 10^5^ ± 2081.7	1.5 × 10^5^ ± 1423.1
	2	1.5 × 10^5^ ± 11,135.5	0
	7	1.5 × 10^5^ ± 14,977.8	0
	14	1.5 × 10^5^ ± 1527.5	0
	28	4.8 × 10^5^ ± 20,816.7	0
*S. aureus* Count CFU/gr	0	1.5 × 10^5^ ± 7937.3	1.5 × 10^5^ ± 11,269.4
	2	1.1 × 10^5^ ± 1527.5	5.3 × 10^1^ ± 6.5
	7	8.6 × 10^2^ ± 15.3	0
	14	3.8 × 10^3^ ± 602.8	0
	28	4.4 × 10^3^ ± 160.4	0
*A. brasiliensis* Count CFU/gr	0	1.5 × 10^5^ ± 14.4	1.5 × 10^5^ ± 13,000.0
	7	1.0 × 10^2^ ± 15.1	6.7 × 10^1^ ± 13.1
	14	0	0
	28	0	0
*C. albicans* Count CFU/gr	0	1.4 × 10^4^ ± 212.1	1.4 × 10^4^ ± 1792.5
	7	1.9 × 10^3^ ± 10,343.2	0
	14	2.5 × 10^3^ ± 136.5	0
	28	1.8 × 10^3^ ± 113.7	0

## Data Availability

Not applicable.

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
