# Peer review of "Biogenic Silver Nanoparticles from Iris tuberosa as Potential Preservative in Cosmetic Products"

_molecules, 2021, doi:10.3390/molecules26154696_

Round 1

Reviewer 1 Report

Dear Authors,

I have the following comments to your work:

  1. please, read your work again and try to correct the writing errors (spelling errors, grammar errors, etc). please, pay attention to the names of the bacterial strains: the species name should be written with small letters, flavonoids, gallic acid, etc. should be written with small letters. Also some sentences are not clear - please check the introduction with a native speaker.
  2. why did the Authors select this particular plant species? please comment on that
  3. what is the composition of the extract used for the preparation of AgNPs?
  4. in the table 3 indicate which data are for gentamicin and which for tebuconazole 
  5. how do the Authors conclude that their bio-AgNs are strong antioxidants? (line 267) please, add a comparison to a reference compounds tested in the same conditions
  6. please add city and country to every producer presented in the materials and methods section and to the Botanical Garden
  7. how come that the Authors added sodium carbonate wih Folin Ciocalteu together? it is suggested to first use F-C reagent and later carbonates to have a better color formation. Can you provide any reference for this type of procedure?

Author Response

Q1.     please, read your work again and try to correct the writing errors (spelling errors, grammar errors, etc). please, pay attention to the names of the bacterial strains: the species name should be written with small letters, flavonoids, gallic acid, etc. should be written with small letters. Also some sentences are not clear - please check the introduction with a native speaker.

A1. The manuscript has been checked by a native speaker and all the errors have been changed as suggested.

Q2. why did the Authors select this particular plant species? please comment on that

A2. Iris tuberosa is a medicinal plant widely used in traditional medicine for its properties due to flavonoids and polyphenol composition. These compounds act as reducing agents and are commonly used in green synthesis of silver nanoparticles. furthermore, this plant has never been used as platform to obtain silver nanoparticles.  Moreover, the culture of this plant does not require any special conditions which can facilitate the possibility to scale-up the production of this extract.

Q3. what is the composition of the extract used for the preparation of AgNPs?

A3. It is known that extracts obtained from different species of Iris spp. contain more than 50 compounds mainly composed by flavonoids, phenols and terpenes (Antibiotics 2020, 9, 403; doi:10.3390/antibiotics9070403). The extract from Iris tuberosa

Q4. in the table 3 indicate which data are for gentamicin and which for tebuconazole 

A4. Data from table 3 have been changed as suggested.

Q5. how do the Authors conclude that their bio-AgNs are strong antioxidants? (line 267) please, add a comparison to a reference compounds tested in the same conditions

A5. DPPH (2,2-diphenyl-1-picryl-hydrazyl-hydrate) free radical method is an antioxidant assay based on electron-transfer that produces a violet solution in ethanol. This free radical, stable at room temperature, is reduced in the presence of an antioxidant molecule, giving rise to colorless ethanol solution. The use of the DPPH assay provides an easy and rapid way to evaluate antioxidants by spectrophotometry, the control reference is usually a known compound with high antioxidant activity as rosmarinic acid, ascorbic acid etc, in our case we used as standard gallic acid which is widely used (http://dx.doi.org/10.1016/j.colsurfb.2012.08.041, https://doi.org/10.3390/nano7100306; https://doi.org/10.1016/j.lwt.2020.109769)

Q6. please add city and country to every producer presented in the materials and methods section and to the Botanical Garden

A6. Done as suggested

Q7. how come that the Authors added sodium carbonate wih Folin Ciocalteu together? it is suggested to first use F-C reagent and later carbonates to have a better color formation. Can you provide any reference for this type of procedure?

A7.  We used the same method described by lin and Tang (Food chemistry 2007, (101), 140-147 https://doi.org/10.1016/j.foodchem.2006.01.014) following the same steps

Reviewer 2 Report

The Authors presented an interesting manuscript concerning the potential of biogenic silver nanoparticles from Iris tuberosa as preservative in cosmetic products. The manuscript contain a lot of information and extensive research however should be improved.

Abstract should be improved.

There is no need to highlight twice who performed the tests in one sentence (“In our study we obtained biogenic-Silver nanoparticles 12 from Iris tuberosa leaves extract”).

The sentence “Antioxidant and antimicrobial properties were evaluated against typical microbial contaminants found in cosmetic products, showing high antioxidant and antimicrobial properties” suggest that antioxidant properties were evaluated against microorganisms. It should be rewritten.

In the last sentence the Authors should present mainly the results of the research, not its purpose.

The Introduction contains the necessary information but English should be corrected.

Line 93 - the sentence starts with a lowercase letter.

The methodology is described in a sufficiently comprehensible way, however the names of microorganisms are written incorrectly (lines 316 and 363).    

The results and discussion is exhaustive, however some parts contain intensive discussion, while in others the discussion is poor. The names of microorganisms are written incorrectly in the text. Table 5 is unreadable and should be transformed.

After the minor correction manuscript could be recommended to publish. 

Author Response

Q1.Abstract should be improved.

A1.The Abstract has been changed as suggested

Q2.There is no need to highlight twice who performed the tests in one sentence (“In our study we obtained biogenic-Silver nanoparticles 12 from Iris tuberosa leaves extract”).

A2.The sentence has been changed by “In our study, biogenic-Silver nanoparticles were obtained from Iris tuberosa leaves extract”

Q3.The sentence “Antioxidant and antimicrobial properties were evaluated against typical microbial contaminants found in cosmetic products, showing high antioxidant and antimicrobial properties” suggest that antioxidant properties were evaluated against microorganisms. It should be rewritten.

A3. Antioxidant has been removed from the sentence and now appear as Antimicrobial properties were evaluated against typical microbial contaminants found in cosmetic products, showing high antimicrobial properties.

Q4. In the last sentence the Authors should present mainly the results of the research, not its purpose.

A4.A new paragraph has been added “xxxxx”

Q5. The Introduction contains the necessary information but English should be corrected.

A5. The manuscript has been checked by a native speaker

Q6. Line 93 - the sentence starts with a lowercase letter.

A6. Done as suggested

Q7. The methodology is described in a sufficiently comprehensible way, however the names of microorganisms are written incorrectly (lines 316 and 363).    

A7. Changed as suggested all over the manuscript

Q8.The results and discussion is exhaustive, however some parts contain intensive discussion, while in others the discussion is poor. The names of microorganisms are written incorrectly in the text. Table 5 is unreadable and should be transformed.

A8. The results and discussion has been rewritten and Table 5 has been changed

Reviewer 3 Report

In this manuscript, the authors developed biogenic-Silver nanoparticles from Iris tuberosa leaves extract, which showed high antioxidant and antimicrobial properties. The results indicate that the Biogenic AgNPs are a promising alternative to use as a cosmetic preservative. Overall, it is an interesting work. However, some questions should be clarified or corrected in the presented manuscript before publication in Molecules. I suggest Minor Revision before acceptance. Some detailed issues are as follows:

  1. It is noted that your manuscript needs to be edited more carefully. There are a lot of grammatical, spelling and formalerrors in the manuscript. Please improve writing skills, including spelling, capitaland small letter, lexical variation and logical connectives between adjacent paragraphs in this manuscript.
  2. The authors should supplement the elemental analysis of AgNPs.

      3. As a cosmetic preservative, the biosafety assessment of Biogenic AgNPs           should supplement.

  1. The manuscript lacks any information on statisticalmethods and experimental replication. This is particularly worrisome. Please revise the manuscript detailing your experimental and technical replications.

Author Response

Q1. It is noted that your manuscript needs to be edited more carefully. There are a lot of grammatical, spelling and formalerrors in the manuscript. Please improve writing skills, including spelling, capitaland small letter, lexical variation and logical connectives between adjacent paragraphs in this manuscript.

A1. The manuscript has been checked by a native speaker and all the errors have been changed as suggested.

Q2. The authors should supplement the elemental analysis of AgNPs.

            Unfortunately, the SEM-EDX is broken and we cannot perform the analysis within the following 4 weeks

Q3.  As a cosmetic preservative, the biosafety assessment of Biogenic AgNPs should supplement.

            A3. Many reviews and articles point pout that Ag nanoparticles obtained from green synthesis are found to be very stable, show sufficient preservation efficacy, did not penetrate normal human skin, when the barrier function of human skin is disrupted, Ag nanoparticles on the skin surface may penetrate the skin. It may be possible that 0.2% to 2% of Ag nanoparticles could penetrate the skin (0.002 - 0.02 ppm). At these levels Ag nanoparticles did not show any toxicity. Ag nanoparticles appear to be suitable for use as a preservative in cosmetics (http://dx.doi.org/10.4236/jcdsa.2016.61007; https://doi.org/10.1016/j.nano.2009.12.002, https://doi.org/10.1016/j.ecoenv.2018.12.095). Furthermore, the Scientific Committee on Consumer Safety of European Commission ruled silver nanoparticles as safety ingredient used in cosmetic products. (S. Committee and C. S. Sccs, Scientific Committee on Consumer Safety Colloidal Silver (nano), no. October. 2018)

Q4. The manuscript lacks any information on statisticalmethods and experimental replication. This is particularly worrisome. Please revise the manuscript detailing your experimental and technical replications.

             A4. Done as suggested

Round 2

Reviewer 1 Report

Dear Authors,

concerning the corrections from the native speaker, i can see no changes made in the revised version of the manuscript that were highlighted in yellow. Please, perform a thorough correction of your work.

please, include a detailed description of the composition of the extract used in the study, as I asked before.

please, explain why the Authors perceive 24-65% of DPPH radical inhibition at the concentration higher than 1 mg/mL as strong (see line 267)? please, support your conclusions with scientific literature otherwise modify your assessment

Author Response

Q1-concerning the corrections from the native speaker, i can see no changes made in the revised version of the manuscript that were highlighted in yellow. Please, perform a thorough correction of your work.

A1-Done as suggested

Q2-please, include a detailed description of the composition of the extract used in the study, as I asked before.

A2-The the antioxidant compounds aqueous plant extract was used as a platform to obtain silver nanoparticles, these metabolites were employed as reducing vehicle to obtain silver in an optimal oxidation state for the synthesis. Thus, in our work we have only focused on determining polyphenols and flavonoids because they have been reported to contribute as precursors to synthetize nanoparticles. However, we cannot rule out that other metabolites such as carotenoids may contribute to this mechanism. A new sentence referred to the content of polyphenols and flavonoids has been added.

Q3-please, explain why the Authors perceive 24-65% of DPPH radical inhibition at the concentration higher than 1 mg/mL as strong (see line 267)? please, support your conclusions with scientific literature otherwise modify your assessment

A3- The referee is right, the antioxidant activity is not that strong as was stated before, we have modified this statement in the manuscript file.